# Research and analysis of an enhanced genetic algorithm identification method based on the LuGre model

**Wanjun Zhang[1,2,3], Feng Zhang[1,2], Jingxuan Zhang[1,2], Siyan Zhang[1,2], Jingyi Zhang[1,2], Jingyan Zhang[1,2], Honghong Sun[4], Kristian E. Waters[5], Hao Ma[4]***

1 Gansu ZeDe Electronic Technology Co., Ltd., Tianshui, China, 2 Gansu Dingxi Technology Co., Ltd., Tianshui, China, 3 Gansu Xionglin Technology Co., Ltd., Tianshui, China, 4 BGRIMM Technology Group, Beijing, China, 5 Department of Mining and Materials Engineering, McGill University, Montreal, Quebec, Canada

* haoma@bgrimm.com

## Abstract

Nonlinear friction in high-precision, ultra-low-speed servo systems severely degrades performance, causing low-speed crawling, static errors, and limit-cycle oscillations. This study introduces the LuGre friction model to describe these phenomena mathematically and proposes an improved genetic algorithm (GA) for precise parameter identification. Simulations demonstrate that LuGre-based feedforward compensation outperforms conventional proportional-integral-derivative (PID) control, effectively mitigating speed tracking errors and enhancing both speed and position accuracy. Experimental validation on a linear motor platform confirms the method's efficacy, achieving a 25.1% improvement in tracking accuracy. The results highlight the practical relevance of this approach for precision servo systems. This work has achieved a practical identification framework for LuGre parameters, combining GA optimization with transient/steady-state data, feedforward compensation that directly injects estimated friction forces, bypassing feedback delays and experimental verification of the method's industrial applicability.

## 1 Introduction

Friction is a critical factor influencing the performance of electromechanical servo systems. To enhance system performance, appropriate friction compensation methods must be employed to mitigate or eliminate the adverse effects of friction [1–3]. It is essential to address nonlinear frictional forces for optimal servo system functionality [4].

Among the various friction compensation methodologies, those based on friction models are particularly effective as they can deliver swift and accurate compensation, making them a widely adopted approach. In terms of reducing the impact of friction, compensating for nonlinear friction is more practical and cheaper than pure

**Data availability statement:** All relevant data are within the paper and its Supporting Information files.

**Funding:** This work is supported by NSFC (51321004, 92475109) and National Science and Technology Major Project (2018ZX040001-181).

**Competing interests:** No authors have competing interests.

mechanical methods, such as improving contact surface materials and lubrication conditions [5–7].

In order to compensate for nonlinear friction, the key step is to establish an accurate friction model and accurately determine the parameters of the model, which is helpful for conducting friction compensation research in precision mechanical system engineering applications, such as trajectory tracking and dynamic prediction. So far, many effective mathematical models have been proposed to describe friction characteristics, such as Coulomb model, Stribeck model, Dahl model, Leuven model, LuGre model, etc [7]. Among them, the LuGre friction model is a relatively complete friction model that can accurately describe the characteristics of viscous sliding motion, friction hysteresis, pre sliding displacement, and variable maximum static friction force during the friction process [8]. During low-speed operation of the motor, negative phenomena such as low-speed crawling, unstable operation, and limit cycles are particularly prone to occur, all of which are caused by nonlinear friction [9,10].

There are two main types of friction compensation methods [11–15]. The first type is compensation that is not based on friction models, treating friction as a disturbance and suppressing its effects by providing the system with anti-interference capabilities.

Zhao et al. [16] studied a novel sliding mode control algorithm with adaptive gain, introduced an integral term to design the sliding surface, and selected a torsion control algorithm based on high-order SMC idea as the switching control law. Ali K et al. [17] proposed an advanced control algorithm that combines fast integral terminal sliding mode control, robust exact differentiator, and feedforward neural network-based estimator. Firstly, considering the dynamic LuGre friction compensation model, a mechanical manual model of an n-degree-of-freedom system was established. Then, a friction compensation based FIT-SMC nonlinear control method was proposed, which showed better performance than traditional compensators. However, it is pointed by author that it is a computationally complicated function that takes a long time to execute.

Another work that Ali K et al. [18] proposes is an integral sliding mode control (ISMC) for an anthropomorphic manipulator using a modified LuGre friction model with friction compensation. His contributions include deriving a mathematical model for a 5-DOF AUTAREP robotic manipulator, incorporating kinematics, dynamics, and the LuGre friction model, designing a robust controller to compensate for dynamic friction and ensure precise trajectory tracking. The ISMC law, based on the Lyapunov function, addresses system parametric and load torque uncertainties, and implementing the control law on a custom platform with LabVIEW and NI myRIO hardware, including motor drive circuits. However, negative aspect in the proposed technique ISMC with friction compensation is that the temperature effects of joints are not considered as well high control gains and high control efforts related to chattering issues.

Furthermore, Ali K et al. [19] proposes a fault-tolerant control (FTC) strategy for a 5-DoF robotic manipulator, combining an observer-based approach with hardware redundancy to enhance performance under actuator and sensor faults. A dynamic LuGre friction model is derived as the basis for control law design. For FTC, an adaptive back-stepping method is used for fault estimation, while nominal control

laws enable controller reconfiguration and observer design. Fault detection compares actual and observed states, followed by fault-tolerant methods using redundant sensors. Results confirm the effectiveness of the proposed FTC strategy with model-based friction compensation, demonstrating improved tracking performance and robustness in the presence of friction and faults. While Mehmood Y et al. [20] proposes fractional and integral order fuzzy sliding mode controllers (FSMC, FFSMC) for skid-steered vehicles (SSV) with variable friction coefficients and displaced center of gravity (CG). The FFSMC reduces forces from ground-tire interaction during skidding and friction variations. Implemented on a four-wheel SSV during high-speed turns, the controllers were tested in a simulation environment integrating SSV dynamics and a wheel-road model. Tests used a pioneer-3AT (P-3AT) robot SSV with displaced CG and variable tire friction. Results show the FFSMC achieves reduced state errors and minimal chattering, compensating for wheel response variations due to CG shifts. Additionally, fuzzy tuning minimizes chattering from conventional sliding mode controllers (SMCs). P. Moradi et al. as well as M. H. Korayem et al. has also performed similar work and achieved significant findings [21–24].

Cong [25] designed an improved Stribeck friction model. Based on the improved model, an inertia stabilization platform and speed stabilization loop feedforward control were designed in the closed-loop control system to avoid vibration and limit cycle problems caused by changes in motion direction and excessive friction compensation. The experimental results show that, with the use of nonlinear friction model compensation, the isolation performance of both the tracking system and the isolation system is superior to the corresponding control system.

Aguilar Avelar et al. [26] first discussed the modeling, identification, and compensation of the influence of nonlinear friction on flywheel control. A new algorithm for stabilizing the flywheel in an unstable position and a friction compensation controller based on feedback linearization have been proposed. The experimental results show that the established asymmetric Coulomb friction element dynamic model can better characterize the real experimental platform of the system, with good stability and low power consumption control performance. However, the velocity of the pendulum s slightly greater for the proposed algorithm.

Li et al. [10] proposed an improved Stribeck friction model (SFM) and an optimization algorithm consistent with the positioning platform, and verified a compensator based on the friction model and disturbance observer through simulation. The simulation results show that the compensator based on friction model with added disturbance has better performance than traditional compensators. In addition, compensation comparisons were made between Coulomb friction, traditional SFM, and improved SFM. The experimental results showed that compared with Coulomb friction compensation, the improved SFM compensation reduced the following errors by 67.67% and 51.63% at speeds of 0.005m/s and 0.05m/s, respectively. However, the proposed modified SFM has a limitation in terms of velocity reversals. Fixed model friction compensation includes a fixed friction model compensation term, which remains unchanged throughout the entire control process as it is a parameter fixed friction model. Therefore, the accuracy requirements for the friction model are extremely high, but in actual systems, factors such as time and temperature affect friction, and fixed friction compensation terms inevitably have errors. For the method of fixed model friction compensation, it not only needs to meet the high requirements for model accuracy, but also needs to have better results in practical situations with extremely low environmental impact.

Han et al. [27] established a robust positioning control scheme using a friction parameter observer and a recursive fuzzy neural network based on sliding mode control. The friction state observer fully captures nonlinear dynamic friction, and uses recursive fuzzy neural network technology to establish an approximation method for system uncertainty, improving positioning accuracy. The performance of the control scheme is verified through experiments. However, for the purpose of more enhancement of precision positioning, the authors consider the robust control using the RFNN estimator and adaptive approximation error estimator.

Meng et al. [28] constructed an adaptive robust controller for dynamic friction compensation based on the LuGre friction compensation model. Adopting online recursive least squares estimation for model parameter identification, achieving real-time parameter updates, and designing a sliding mode controller to alleviate the degradation of control performance

caused by parameter estimation errors and unmodeled dynamics. It is noted that since the trajectory is not always persistently exciting, the parameter estimates exhibit slow convergence. In Jin et al.'s work [29], a complementary sliding mode control method based on approximation angle saturation function is proposed, which can reduce the boundary layer as the state trajectory changes until it converges to the sliding surface. This method effectively eliminates system chattering and improves position tracking accuracy and robustness. The designed control method was compared with the controller using traditional saturation function, and the results showed that the control method can achieve lower tracking error and stronger robustness. However, it can only be applied to the case where the trajectory of the state moves towards the switching surface, also the friction force is considered as a part of the lumped uncertainty, and the influence of friction force on the system is not properly analyzed.

The second type of friction compensation methods is error compensation based on specific friction models. This type of compensation method relies on establishing mathematical models that can accurately reflect various frictions in the system.

Lu et al. [30] proposed a sliding mode controller based on the approach law for controlling the position of permanent magnet synchronous motors. In addition, an adaptive disturbance observer was designed to suppress external disturbances and parameter uncertainties, and the designed controller was validated through experiments.

Chen et al. [31] proposed a discrete adaptive sliding mode controller for high-precision motion control of linear permanent magnet synchronous motors, which achieves real-time gain tuning capability and robustness, ensuring the performance of tracking tasks. Experimental results show that the designed controller has fast response and strong robustness to uncertainty and noise. However, maximum static friction force may arouse a higher tracking error at the beginning of the move.

Zhao et al. [32] proposed an adaptive neural network nonsingular fast terminal sliding mode control for high-precision position tracking of permanent magnet synchronous motors in the presence of uncertainty. Nonsingular fast terminal sliding mode control was used to achieve rapid convergence of the system at the equilibrium point. However, the approach often inadequately addresses parameter uncertainty, compromising the quality of identification outcomes

Currently, mainstream identification algorithms for LuGre friction model parameters include GAs, particle swarm optimization algorithms, and neural network algorithms. Despite the plethora of methodologies available for friction identification and compensation, challenges persist regarding parameter identification accuracy and chattering phenomena during feedforward compensation processes at velocity-zero positions [33].

To enhance speed and position tracking accuracy in high-precision, ultra-low-speed servo systems, an improved LuGre GA is employed for identification and compensation, achieving precise control objectives for the platform. Nonlinear friction links significantly impact the dynamic and static performance of high-precision, ultra-low-speed servo systems, largely evidenced by low-speed crawling, substantial static errors, or steady-state limit cycles. This work introduces the fundamental principles and mathematical expressions of the LuGre friction model and presents an advanced GA identification method based on LuGre. Simulation analyses demonstrate that, compared to traditional PID control, incorporating feedforward friction compensation based on the LuGre model effectively mitigates speed tracking discrepancies attributed to nonlinear friction, thereby enhancing both speed and position tracking accuracy while offering greater engineering practicality. In fact, these intelligent algorithms also have some shortcomings, such as poor generalization performance of NNA when processing large amounts of data, slow convergence speed of SAA, and low accuracy and easy divergence of PSO algorithm. Considering the multi parameter characteristics of the friction model established by the research object, this paper intends to use a genetic algorithm with relatively ideal parallel iteration ability as the identification method for the friction model.

The manuscript is composed of five sections. Section 1 introduces the LuGre friction model, detailing its fundamental principles and mathematical formulation. The LuGre model comprehensively describes both the dynamic and static characteristics of friction in servo systems. Accurate identification of the model parameters is crucial for achieving effective

friction compensation. Section 2 proposes an improved genetic algorithm-based identification method for the LuGre model. This section primarily presents a detailed description of the LuGre friction model and the identification framework. Section 3 elaborates on the theoretical model of the enhanced genetic algorithm identification method proposed in this paper for the LuGre model. Section 4 constructs a linear motor platform and outlines its configuration and parameters. It provides a detailed explanation of the parameter identification method for the LuGre friction compensation model and validates the effectiveness of the friction compensation technique as well as the robustness of the designed controller on the physical platform. The performance of the proposed parameter identification algorithm is verified through simulation examples and compared with existing methods. Section 5 concludes the paper and discusses potential future work.

The main contributions of this paper can be summarized: A novel parameter identification algorithm is proposed by integrating an adaptive evolution module and an optimal solution precision optimization module into the traditional genetic algorithm; Based on simulation experiments, it is demonstrated that the proposed algorithm can effectively identify the six parameters of the LuGre friction model; To highlight the advantages of the proposed method, it is compared with existing approaches. The results show that the method significantly improves the accuracy of parameter identification while ensuring fast convergence. Simulation results validate the effectiveness and feasibility of the algorithm.

## 2 Identification structure model

### 2.1 Traditional friction model

Currently, friction feedforward compensation in servo feed systems predominantly relies on the traditional Stribeck friction model, as illustrated in Fig 1. This model, which reflects the characteristic behavior of friction on lubricated metal surfaces, can be categorized into four distinct stages. As the operational speed progressively increases, these stages encompass static friction, boundary lubrication friction, partial fluid lubrication friction, and full fluid lubrication friction. Notably, under conditions of partial fluid lubrication, an increase in system operating speed results in a decrease in the frictional force between the two metal contact surfaces. This phenomenon is referred to as the Stribeck effect.

Traditional friction model expression:

$$g\left(\dot{x}(t)\right) = F_c + (F_s - F_c) \cdot e^{-\left(\frac{\dot{x}(t)}{v_s}\right)^2} \tag{1}$$

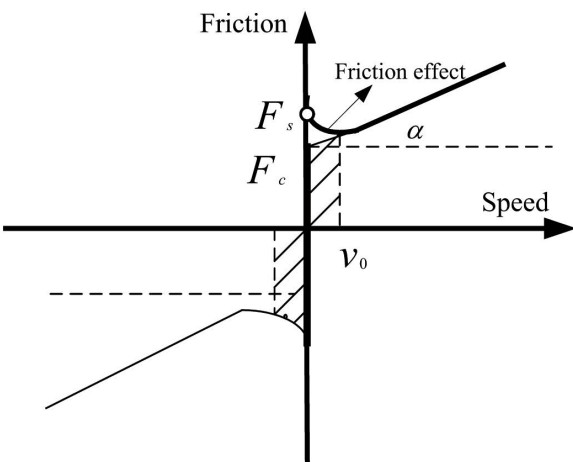

**Fig 1. Traditional Stribeck friction model.**

In Equation (1), $\dot{x}(t)$ represents the relative velocity between the contact surfaces; $v$ represents the current motion speed of the system; $\alpha$ is the coefficient of viscous friction; $t$ refers to time, $v_s$ denotes the Stribeck speed; $F_c$ refers to the Coulomb friction force; and $F_s$ is defined as the maximum static friction force. Consequently, this defines the static coefficient.

As illustrated in Fig 1, the LuGre friction model introduced by Canudas et al. [12] conceptualizes the interaction between two contacting surfaces as comprising numerous fine elastic asperities. When there is relative displacement between the two objects, these asperities experience elastic deformation, resulting in a restoring force that manifests as frictional force. In details, this model posits that the contact between rigid body surfaces is mediated by elastic bristles, with the lower surface material exhibiting greater stiffness compared to the upper surface. When an external force is applied, the bristles undergo deformation due to the tangential force, generating frictional resistance. If the tangential force exceeds a critical threshold, the bristles experience further deformation, leading to sliding motion.

## 2.2 LuGre friction model

The enhanced LuGre genetic algorithm identification method is primarily employed for parameter identification within the LuGre model, encompassing both static and dynamic parameters.

(1) Static parameters [9]: This category comprises static friction, Coulomb friction, Stribeck friction, and viscous friction. These parameters are utilized to characterize the relationship between speed and friction (or friction torque) during steady-state conditions.

(2) Dynamic parameters [10]: Encompassing two parameters that influence dynamic friction response, this set is employed to delineate the dynamic attributes of friction, including sliding friction, pre-sliding displacement, Dahl effect, and Stribeck effect.

The choice of a linear motor for evaluating a nonlinear friction model is driven by its ability to operate at ultra-low speeds, its sensitivity to friction effects, and its suitability for high-precision applications. These characteristics make linear motors an excellent platform for studying and validating advanced friction models like the LuGre model. The LuGre friction model is a comprehensive and widely used dynamic friction model that captures the complex behavior of friction in mechanical systems, it is a dynamic friction model that integrates the Dahl model with steady-state friction characteristics to comprehensively describe friction phenomena such as the Stribeck effect, presliding displacement, hysteresis, and frictional lag. By incorporating the Stribeck effect, this model captures the variable friction behavior that occurs at extremely low velocities, enhancing its ability to simulate real-world friction dynamics.

The average elastic deformation of the asperities is denoted as the state variable $z(t)$, which is utilized in formulating the LuGre friction model.

The model defines friction force $F_f(t)$ as a combination of bristle deflection dynamics and velocity-dependent terms.

1. Friction force:

$$F_f(t) = \sigma_0 \cdot z(t) + \sigma_1 \cdot \frac{dz(t)}{d(t)} + \sigma_2 \cdot \dot{x}(t)$$

(2)

- $\sigma_0$: Bristle stiffness coefficient (N/m).

- $\sigma_1$: Bristle damping coefficient (N·s/m).

- $\sigma_2$: Viscous friction coefficient (N·s/m).

- $z(t)$: Average bristle deflection (m).

- $\dot{x}(t)$: Relative velocity (m/s).

2. Bristle deflection dynamics:

$$\frac{dz(t)}{d(t)} = \dot{x}(t) - \frac{\sigma_0}{g\left(\dot{x}(t)\right)} z(t) \cdot \left|\dot{x}(t)\right|$$

(3)

The function $g\left(\dot{x}(t)\right)$ models the Stribeck effect:

$$g\left(\dot{x}(t)\right) = F_c + (F_s - F_c) \cdot e^{-\left(\frac{\dot{x}(t)}{v_s}\right)^2}$$

(4)

- $F_c$: Coulomb friction (N).

- $F_s$: Static friction (N).

- $v_s$: Stribeck velocity (m/s).

According to the LuGre friction model, when the system reaches a steady state, $\frac{dz}{dt} = 0$:

$$z_s = \frac{g\left(\dot{x}(t)\right)}{\sigma_0} \cdot \frac{\dot{x}(t)}{\left|\dot{x}(t)\right|} = \frac{g\left(\dot{x}(t)\right)}{\sigma_0} \cdot sgn\left(\dot{x}(t)\right)$$

(5)

Where, $sgn\left(\dot{x}(t)\right)$ is a symbolic function, which is defined as:

$$sgn\left(\dot{x}(t)\right) = \begin{cases} -1 & (x < 0); \\ 0 & (x = 0); \\ 1 & (x > 0). \end{cases}$$

(6)

Under steady-speed conditions, the LuGre model represents an exponential model and can effectively characterize the Stribeck phenomenon. Additionally, this model is capable of illustrating phenomena such as friction lag and augmented static friction. At low speeds, the model can capture the Dahl effect. The expression for the friction force $F_r$ during steady-state motion varies with the speed change, and is as follows:

$$F_r = \sigma_0 \cdot z_s + \sigma_2 \cdot \dot{x}(t) = g\left(\dot{x}(t)\right) \cdot z_s + \sigma_2 \cdot \dot{x}(t)$$
$$= \left[F_c + (F_s - F_c) \cdot e^{-\left(\frac{\dot{x}(t)}{v_s}\right)^2}\right] \cdot sgn\left(\dot{x}(t)\right) + \sigma_2 \cdot \dot{x}(t)$$

(7)

### 2.3 Static and dynamic friction to be identified

**(a) Identification of static friction parameters.** According to the Equation ([7]), the parameter vector to be identified is set as follows:

$$\omega_1 = [\hat{F}_c \ \hat{F}_s \ \hat{v}_s \ \hat{\sigma}_2]$$

(8)

In Equation ([8]), $\omega_1$ is static friction parameter vector to be identified.

From the equation above, it can be seen that the LuGre friction model is clear and concise. Compared with other models, this model can smoothly transition from the current friction state to another friction state and has good continuity to a certain extent. In addition, this model can accurately express complex nonlinear friction characteristics such as Coulomb

friction, viscous friction, friction hysteresis, Stribeck effect, and friction memory. The unknown parameters can be obtained through parameter identification methods, providing possibilities for the design and application of various friction compensation algorithms. The relevant parameters that need to be set in the genetic algorithm will be provided in the simulation section later. Combining the above derived statements, the static friction parameters of the LuGre friction model can be obtained.

**(b) Identification of dynamic friction parameters.** The optimal result obtained by identifying the static friction parameters will serve as the basis for identifying the dynamic friction parameters. Dynamic friction parameter identification can be carried out using the acceleration and control force output by the system. The parameter vector to be identified is set as follows:

$$\omega_2 = [\hat{\sigma}_0 \ \ \hat{\sigma}_1]$$

(9)

In Equation (9), $\omega_2$ is Dynamic friction parameter vector to be identified. Combining the LuGre friction model formula, there are:

$$z = \dot{x}(t) - \sigma_0 \cdot \frac{\dot{x}(t)}{g(\dot{x}(t))} z$$

(10)

$$F_r = \hat{\sigma}_0 \cdot z + \hat{\sigma}_1 \cdot \dot{z} + \sigma_2 \cdot \dot{x}(t)$$

(11)

The relevant parameters that need to be set in the genetic algorithm will be provided in the simulation section later. By combining the above formula, the dynamic friction parameters of the LuGre friction model can be obtained.

## 3 Improved annealing genetic algorithm

In order to realize the model and program design, the LuGre friction model of Eq. (12) is discretized:

$$\begin{cases} z(k+1) = \left[\dot{x}(k) - \frac{\sigma_0}{g(k)} z(k) \cdot |\dot{x}(k)|\right] \cdot \Delta T + z(k) \\ g(k) = F_c + (F_s - F_c) \cdot e^{-\left(\frac{\dot{x}(k)}{v_s}\right)^2} \\ F_f(k) = \sigma_0 \cdot z(k) + \sigma_1 \cdot \frac{z(k+1)-z(k)}{\Delta T} + \sigma_2 \cdot \dot{x}(k) \end{cases}$$

(12)

In Equation (12), k represents current sampling time. Initial condition: $z(0) = 0$; $\dot{z}(0) = 0$ enter the corresponding speed $\dot{x}(k)$ to obtain LuGre friction.

Equation (12) includes the state variable $z_s$, which allows for a more comprehensive representation of the frictional process. In friction modeling, the precision of the parameters significantly influences the overall accuracy of the friction analysis. To enhance parameter identification, a parameter identification matrix $\omega = \left(\begin{bmatrix} \sigma_0 & \sigma_1 & \sigma_2 & v_s & F_s \end{bmatrix}, t\right)$ is established, utilizing an advanced identification algorithm. The variance of the deviation between the measured trajectory and the reference trajectory is employed as the evaluation function, as outlined below:

$$e_\omega = \int_t \omega_0(t) - \omega \left(\begin{bmatrix} \sigma_0 & \sigma_1 & \sigma_2 & v_s & F_s \end{bmatrix}, t\right)^2 dt$$

(13)

$$e(k) = F_{fr}(k) - \hat{F}_{fl}\left(\hat{\vartheta}, \dot{x}(k)\right)$$

(14)

In Equation (13) and (14), $u(k)$ is the input of the system; $F_{fr}(k)$ is the actual observed value of the system output, and it represents the LuGre target friction force; $\hat{F}_{fl}\left(\hat{\vartheta}, \dot{x}(k)\right)$ denotes the estimated model calculation value, which is the LuGre friction force calculated from the identified parameters during the iterative process; $e(k)$ is the estimation error.

The fundamental principles of system identification are illustrated in Fig 2.

The definition objective function is:

$$J_l = \frac{1}{2}\sum_{i=1}^{N} e^2(k)(l = 1, 2, \cdots, n)$$

(15)

In Equation (15), $N$ represents the number of samples, with $J_l$ as the identification objective aimed at minimizing the adaptation value denoted as $J$. Here, e signifies the identification error, and Equation (15) serves as the derived objective function. The LuGre model parameter identification flow chart is given in Fig 3.

Comparison curve of objective function (J) is shown in Fig 4.

It is noticeable to point out the effectiveness of the improved genetic algorithm compared with the traditional algorithm; The convergence process of the Genetic Algorithm for LuGre parameter identification is shown in Fig 5.

From Fig 5, it can be clearly observed that in the initial stages of the genetic process, the convergence speed is relatively fast. As the number of generations increases, the convergence tends to get trapped in a local optimum, slowing down the speed. When the number of generations reaches a certain level, the Genetic Algorithm escapes the local optimum and gradually converges to the global optimum. Comparison curve of objective function ($J_k$) is shown in Fig 6.

The comparative results among the traditional genetic identification algorithm, the adaptive genetic identification algorithm, and the algorithm proposed in this study, along with the corresponding objective function value curves, are illustrated in Fig 7.

Figs 6 and 7 present a comparison of the traditional genetic identification algorithm, the adaptive genetic identification algorithm, and the algorithm proposed in this paper, along with the corresponding objective function value curves. The results demonstrate that the identified friction force closely aligns with the model, exhibiting minimal error. However, in cases where the identified maximum static friction force significantly exceeds the model's value, this discrepancy may be attributed to testing errors in the provided data. Although some data points fall outside the expected range, they have not been excluded from the analysis. In comparison to the other algorithms, the proposed algorithm in this study achieves faster and more accurate identification of the friction model parameters.

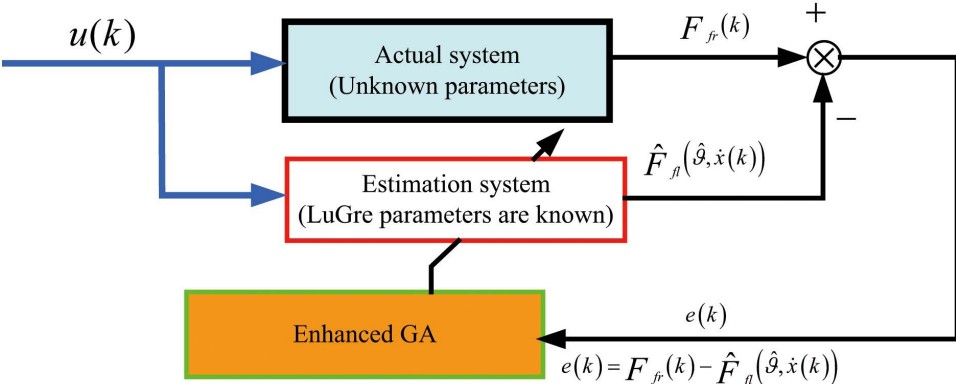

**Fig 2. Basic principles of system identification.**

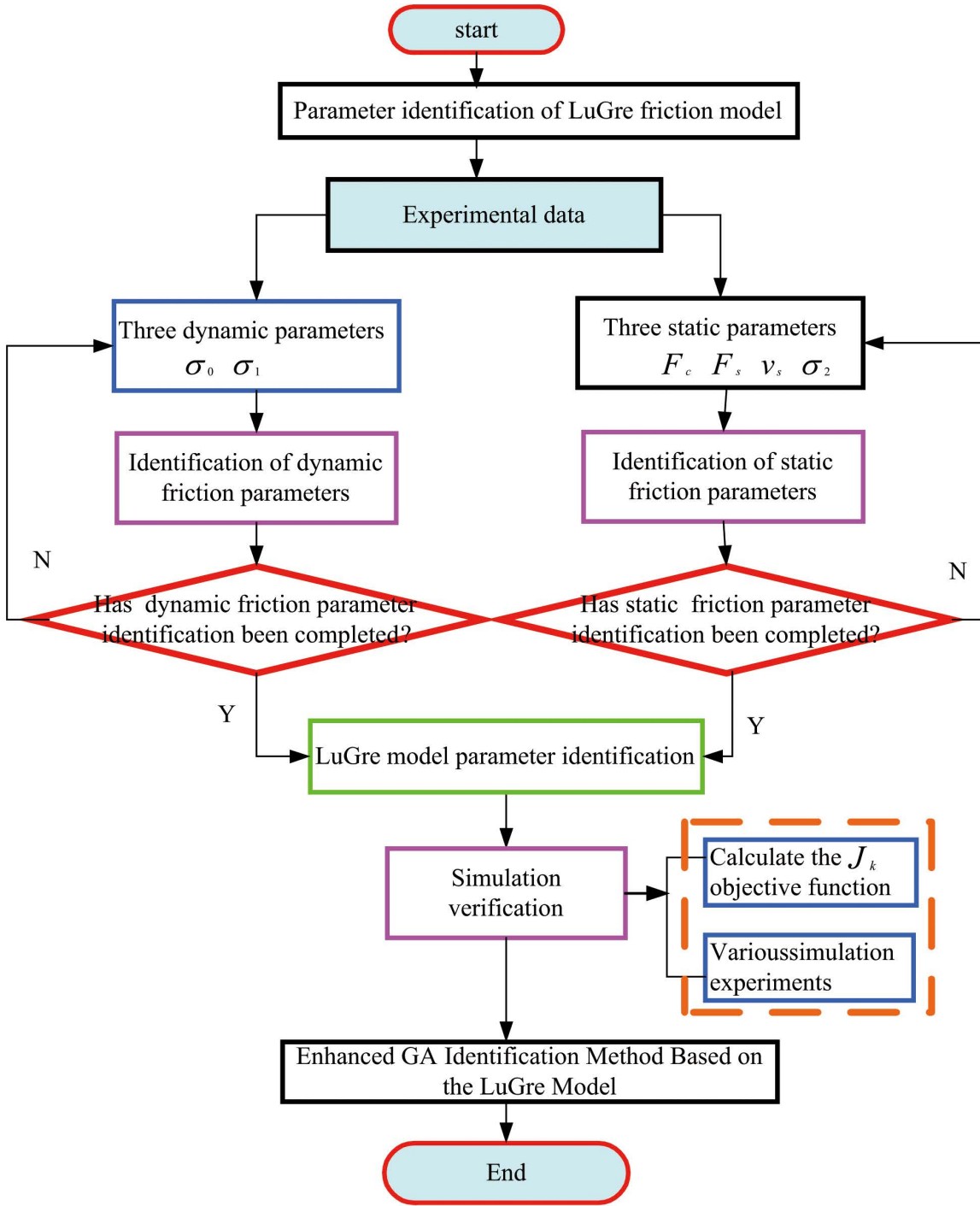

**Fig 3. LuGre model parameter identification flow chart.**

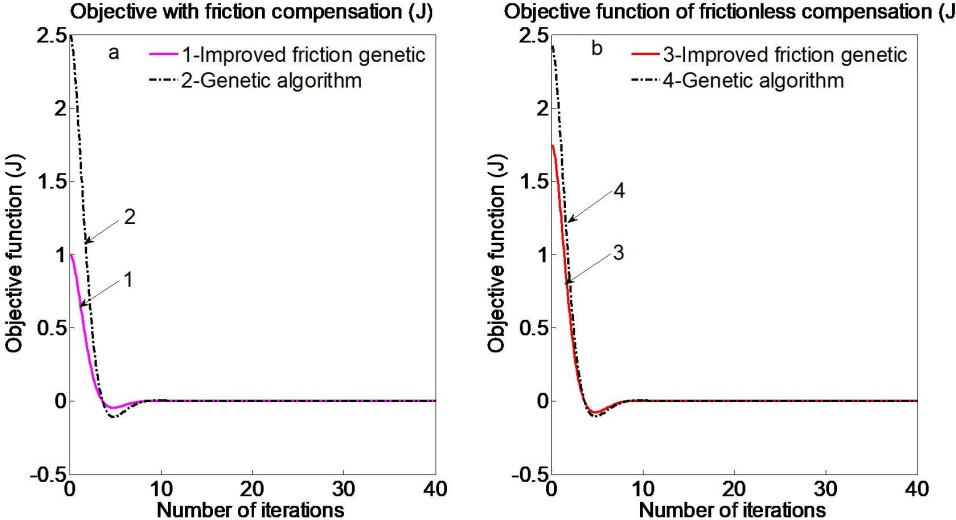

**Fig 4. Comparison curve of objective** function (J).

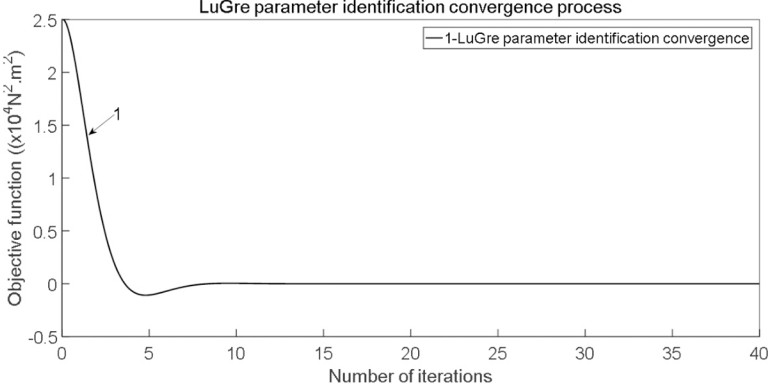

**Fig 5. LuGre parameter identification convergence process.**

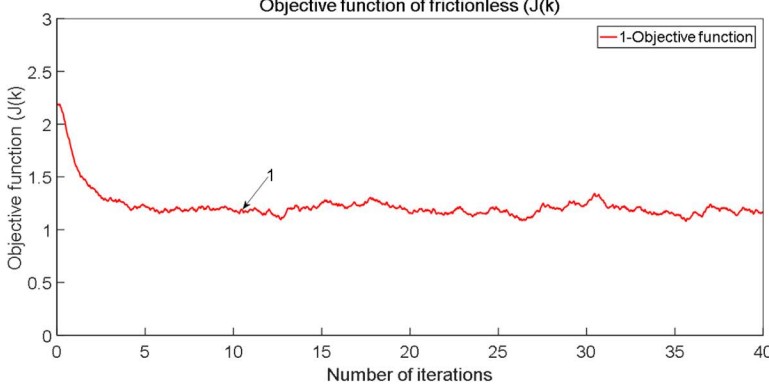

**Fig 6. Comparison curve of objective function ($J_k$).**

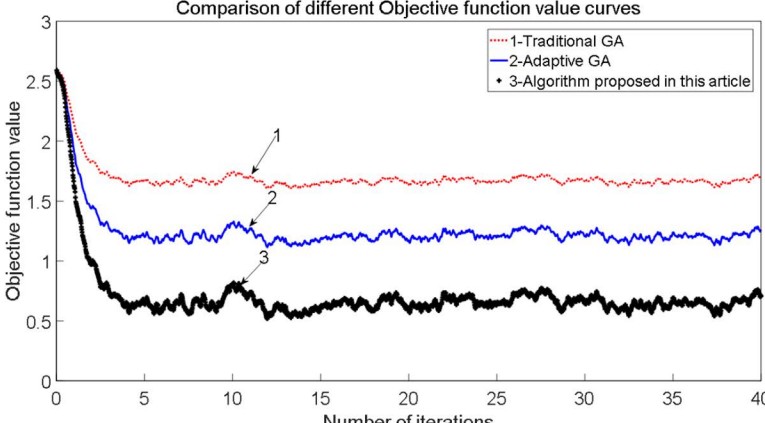

**Fig 7. Comparison of different objective function value curves.**

## 4 Annealing genetic synthesis algorithm based on improved LuGre friction model

The objective of parameter identification and compensation is to enhance the control performance of mechanical systems. Parameter identification involves estimating system parameters, including friction coefficients and dynamic characteristics, through experimental means and data analysis. By accurately determining these parameters, a more refined model of the mechanical system can be developed, thereby facilitating the design and performance optimization of control algorithms.

Parameter compensation entails the development of corresponding compensation algorithms based on the estimated parameter values to mitigate the effects of friction and other nonlinearities within the system. By addressing frictional influences, the tracking error of the system can be minimized, leading to improved control accuracy and stability. Furthermore, parameter compensation allows for adaptation to variations in system parameters, thereby increasing the robustness and adaptability of the control system.

**Step 1**: Define the system parameters and population parameters for the friction model. Specify the population size: $\{x_k\}_{k=1}^{\omega}$ denotes both the upper and lower limits of the population, while the identification vector is represented as $x_k$.

**Step 2**: Implement adaptive calculations for the crossover rate and mutation rate. The crossover rate and mutation rate proposed in this study employ an adaptive strategy. The expressions for the crossover rate and mutation rate are as follows:

$$p_c = \begin{cases} p_{c\min} + (p_{c\min} - p_{c\min}) \cdot \left(\frac{f'}{f_{avg}}\right), & if \quad f' \geq f_{avg} \\ p_{c\max} & , \quad if \quad f' < f_{avg} \end{cases} \tag{16}$$

$$p_m = \begin{cases} p_{m\min} + (p_{m\max} - p_{m\min}) \cdot \left(\frac{f'}{f_{avg}}\right), & if \quad f < f_{avg} \\ p_{m\max} & , \quad if \quad f \geq f_{avg} \end{cases} \tag{17}$$

In Equations (16) to (17):

$p_c$ denotes the crossover variation rate;

$p_{c\min}$ and $p_{c\max}$ represents the minimum and maximum values of the crossover rate, defined as constant parameters;

$p_{m\,\min}$ and $p_{m\,\max}$ indicates the minimum and maximum values of the mutation rate, which are also treated as constant parameters;

$f$ refers to the higher fitness value of the two individuals involved in the crossover;

$f_{avg}$ signifies the average fitness of each generation, as well as the average fitness value of the offspring.

**Step 3**: Implement crossover and mutation operations to generate the next generation population, During the parental crossover process, the participating parental individuals are denoted as $x_i^{l+1}(k)$ and $x_{i+1}^{l+1}(k)$. If the objective function value of $x_i^{l+1}(k)$ exceeds that of $x_{i+1}^{l+1}(k)$, the resultant offspring individuals can be generated according to the following formula:

$$\left.\begin{array}{l} x_i^{l+1}(k) = \xi \cdot x_i(k) + (1-\xi)x_i(k+1) \\ \xi = (\xi_{11}, \xi_{12}, \cdots, \xi_{1n}) \end{array}\right\} \tag{18}$$

In Equation (18): $n$ represents the dimensionality of the variable, while parameter $\xi$ can either be a constant or a random number that follows a uniform distribution.

$$\left.\begin{array}{l} x_i^{l+1}(k) = \eta \cdot x_i(k) + (1-\eta)x_i(k+1) \\ \eta = (\eta_{11}, \eta_{12}, \cdots, \eta_{1n}) \end{array}\right\} \tag{19}$$

In Equation (19): Parameter $\eta$ is defined as a constant or a random number $0 \le \eta \le 1$ that adheres to a uniform distribution. The mutation method employs single-point mutation. It is assumed that the mutation occurs in individual $x_k$, resulting in the new individual as follows:

$$x_i(k) = x_0 \cdot \left(1 + \frac{\tan\left(rand(0,1) - 0.5\right)\pi}{2}\right) \qquad k = 1, 2, \cdots, N \tag{20}$$

**Step 4**: Simulated annealing for new species, initially, establish the number of optimal individuals as $N\_Best$. The fitness experience of the first $N\_Best$ in each generation is unconditionally retained for the next generation, while each subsequent individual is saved with a certain probability following the Boltzmann acceptance criterion. The probability is given by:

$$R_i(x) = \begin{cases} 1 & f_1 \ge f_{N-Best} \\ e^{\frac{-(f_1 - f_{N-Best})}{T_e} \cdot x} & f_1 < f_{N-Best} \end{cases} \tag{21}$$

In Equation (21)

$T_e$ represents the current temperature; $f$ denotes the fitness of the current individual;

$f_{N-Best}$ signifies the individual adaptation with the fitness ranking $N\_Best$.

Consequently, the individual fitness function can be chosen as:

$$\left.\begin{array}{l} G_{\max} = \max_M (J_m) \\ f_M = G_{\max} - J_{\max}\ or\ \frac{1}{J_m} \end{array}\right\} (M = 1, 2, \cdots, n) \tag{22}$$

**Step 5**: Assess whether the current iteration of the evolutionary process has concluded. If not, employ a ranking-based selection method to identify the population with the highest fitness value in the current generation, designating it as the new parent population. Return to Step 3. If the iteration is complete, proceed to Step 6.

**Step 6**: Extract and store the optimal solution derived from the single-round evolutionary process.

**Step 7**: Evaluate whether the predefined number of evolutionary rounds has been achieved. If not, revert to Step 2. If completed, advance to Step 8.

**Step 8**: Compute the fitness value of the optimal solution from each evolutionary round and select the global optimal solution for final output.

**Step 9**: Develop a genetic algorithm tailored to enhance the accuracy and performance of the LuGre friction model (Fig 8).

## 5 Model of friction compensation cross coupling

Perform LuGre genetic algorithm identification testing on a high-precision open servo platform. The structural configuration of the high-precision open servo platform system is illustrated in Fig 9. This platform comprises an industrial control computer, a programmable multi-axis controller (PMAC), servo drives, servo motors, ball screw-nut pairs, linear guides, and a mobile platform.

Physical picture of linear motor platform is shown in Fig 10.

The gantry linear motor platform is manufactured by Akribis company. The X-axis motor has a rated thrust of 221N and a peak thrust of 1248N, with a linear motor constant of 22.4N/W, peak current of 13A, thrust constant of 96N/Arms, and a load mass of 4.6 kg. The linear motor servo drive is a high-order AC servo drive direct drive co-produced by Akribis and Servotronix, featuring a maximum input control voltage of 10V and an instantaneous maximum output control current of 18Arms. Communication with industrial control computer control systems is facilitated through the EtherCAT bus protocol. The position measurement device utilizes Renishaw's absolute grating ruler with a resolution of 50nm.

The industrial control computer is integrated with TwinCAT3 software, responsible for programming control algorithms and real-time calculation of control signals that are sent to the driver. Real-time monitoring of system parameters such as motor displacement, speed, control signal, tracking error, etc., is enabled for easy controller debugging. The closed-loop control system is achieved using the Control System Industrial Control Computer TwinCAT3.

The simulation model for frictional compensation cross coupling is essential for accurately representing and compensating for frictional forces, particularly when constructing nonlinear friction models, conducting parameter identification, and exploring control strategies. The specific research approach of the simulation system for frictional compensation cross coupling is illustrated in Fig. 11 as the simulation and experimental research route.

The curve depicting the relationship between friction and velocity is illustrated in Fig 12.

Sinusoidal signal simulation is shown in Fig 13.

When employing an improved genetic algorithm for the identification of LuGre model parameters, static parameter identification adheres to traditional methodologies. However, for dynamic parameter identification, the displacement (or acceleration) output from the servo system, along with the control shear force, are directly utilized for parameter estimation. In this identification process, the control force serves as the approximation value for the objective function, allowing for the identification of two dynamic parameters. The use of Simulink for simulating the LuGre friction model offers convenience and efficiency. Based on the results obtained from parameter identification, the LuGre friction model leveraging the improved genetic algorithm demonstrates high accuracy in both static and dynamic nonlinear parameter identification, rapid parameter identification speed, and strong robustness performance. Additionally, load change estimations are depicted in Fig 14, while the friction compensation simulation is illustrated in Fig 15.

From Figs 14 and 15, it is evident that the improved LuGre genetic algorithm identification control strategy introduced in this study has significantly enhanced the friction compensation effect compared to traditional methods. Notably, a maximum improvement of 25.1% in tracking accuracy at corners has been achieved. Furthermore, due to the impact of friction, the linear motor may experience deviations when executing feed rate commands issued by the driver. By implementing compensatory controls, these deviations can be minimized, the feed rate error has been reduced from 2.8 m/min to 2.3

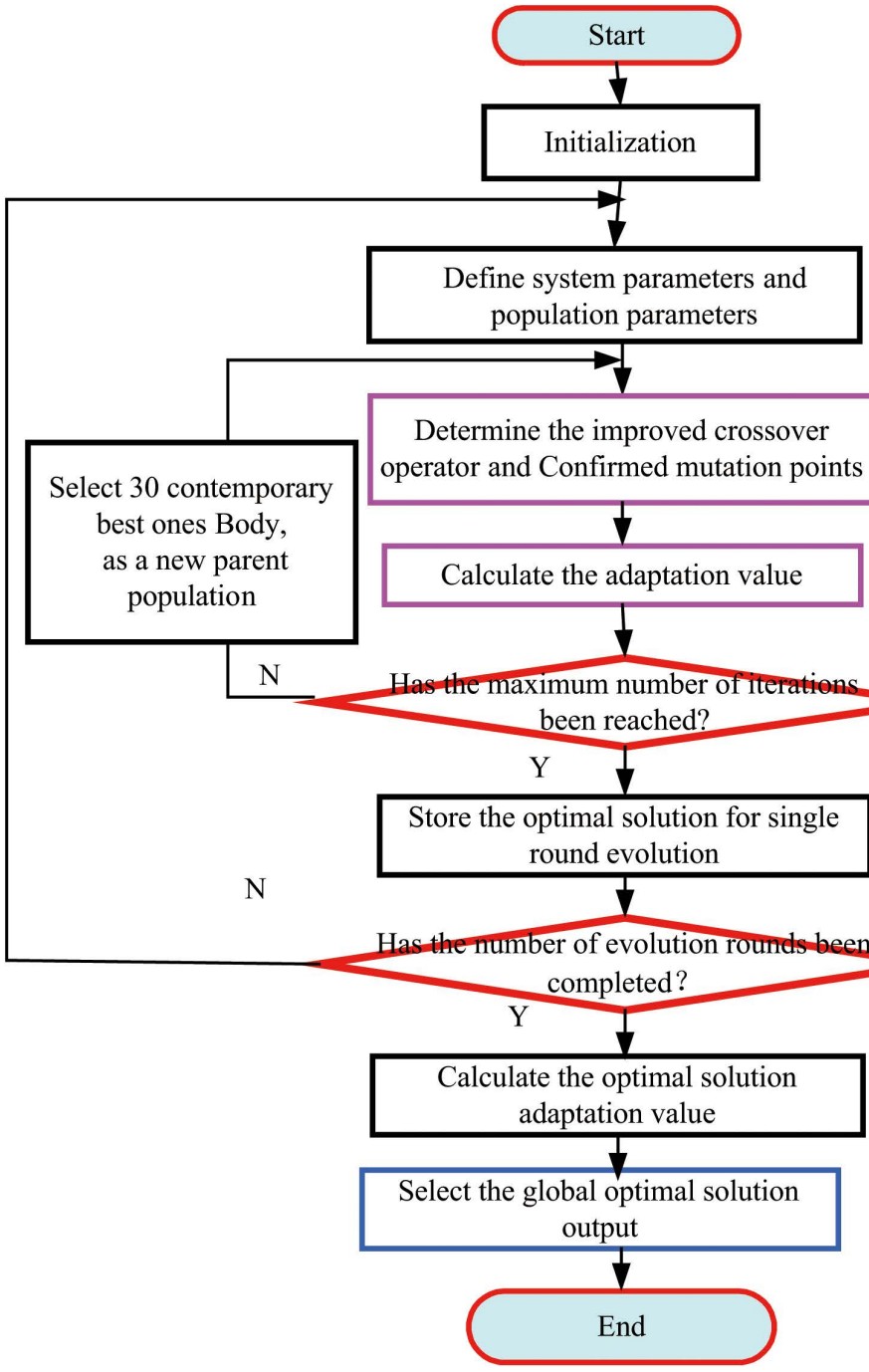

**Fig 8. Flow chart of improved annealing genetic synthesis algorithm.**

m/min, representing a decrease of up to 17.8%. Investigating friction disturbances contributes to a deeper understanding of motion tracking accuracy in linear motors. The improved LuGre genetic algorithm facilitates the identification of friction feedforward compensation strategies, thereby enhancing tracking accuracy and establishing a foundation for improving the overall contour accuracy of precision machining platforms.

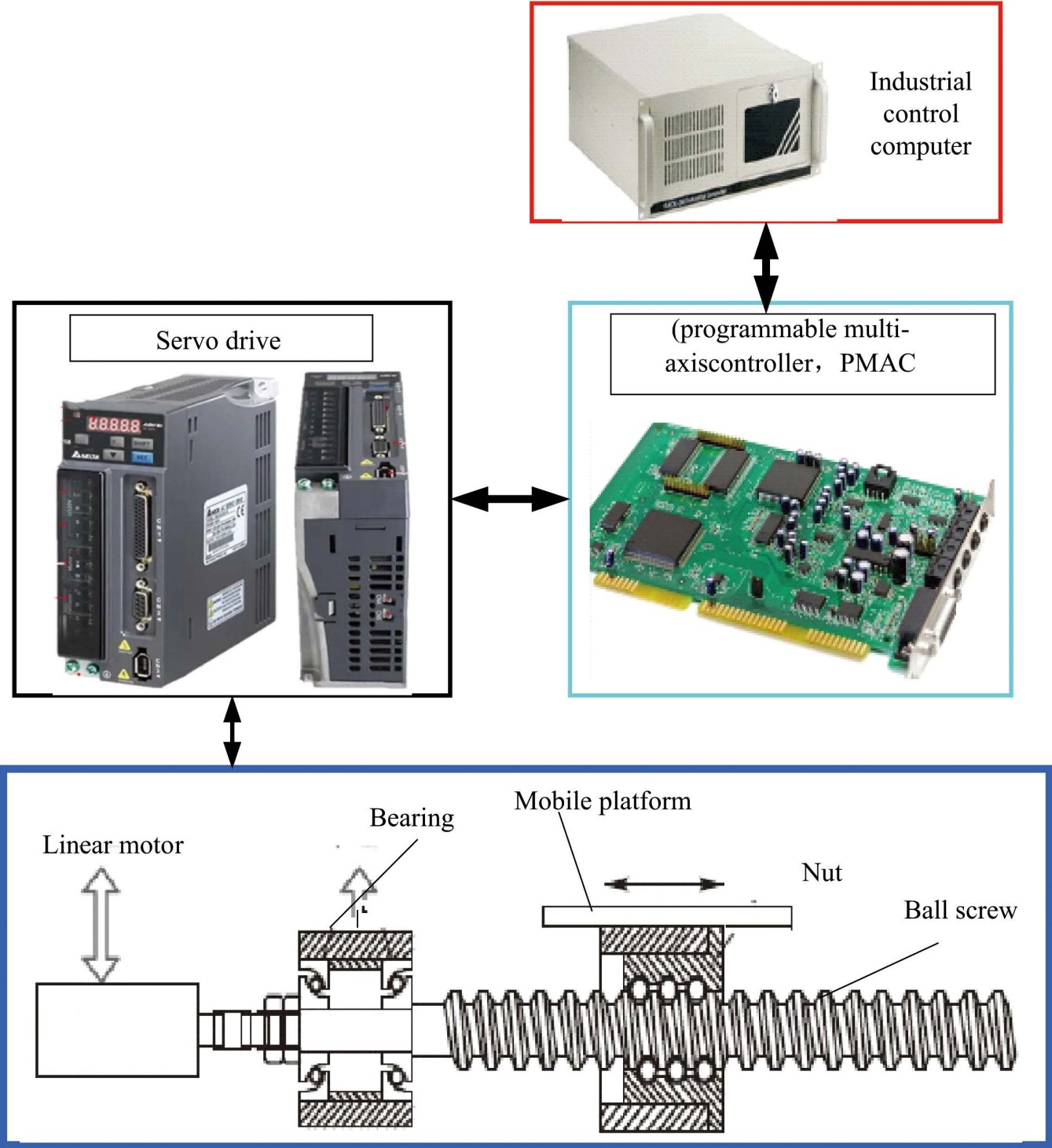

**Fig 9. High precision open servo platform system structure.**

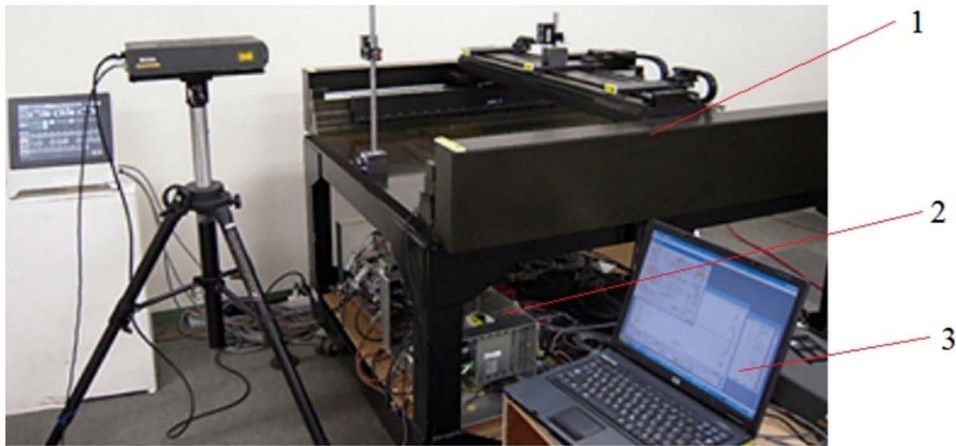

**(a) Linear motor precision motion control experimental platform**

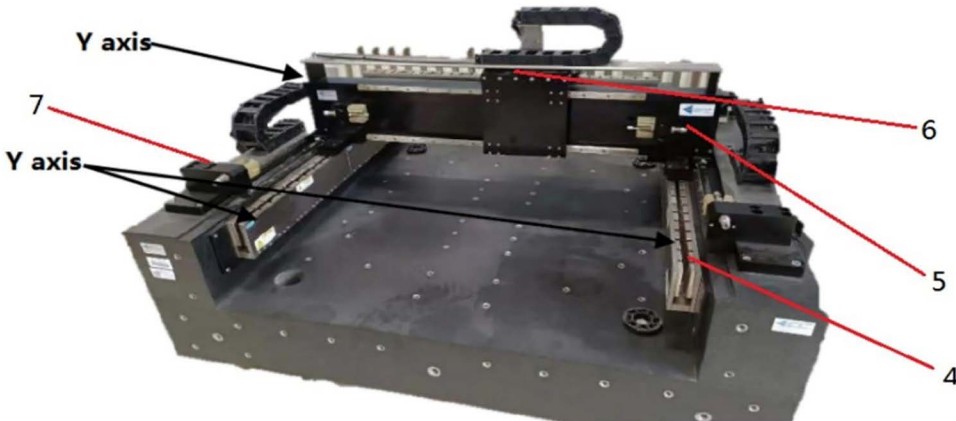

**(b) Composition of Longmen linear motor platform**

**Fig 10. Physical picture of linear motor platform: 1-Gantry linear motor platform; 2-Linear motor servo drive; 3- industrial control computer.; 4-X-axis grating ruler displacement sensor; 5-X-axis motor; 6- industrial control computer; 7- Y-axis grating ruler displacement sensor.**

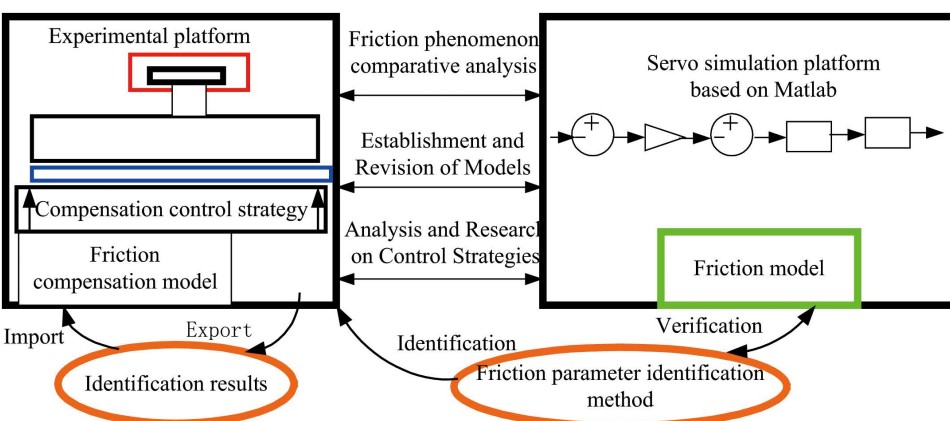

**Fig 11. Simulation and experimental research route.**

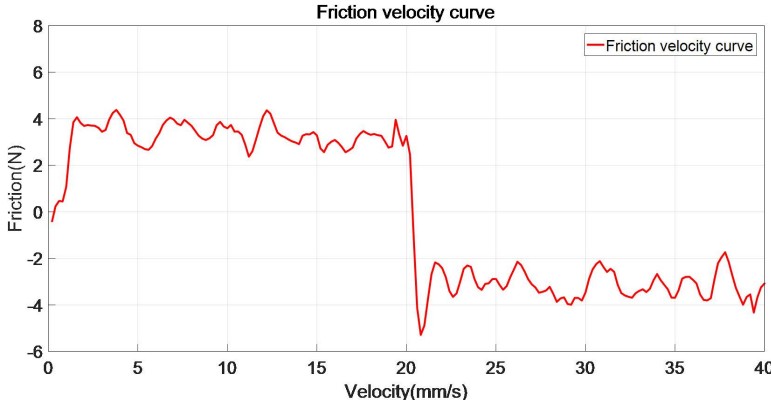

**Fig 12. Friction-velocity relationship curve.**

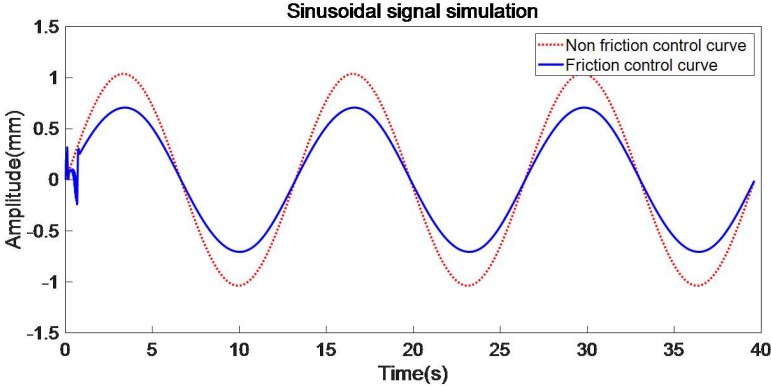

**Fig 13. Sinusoidal signal simulation.**

It is important to note that the offline identification compensation method is more applicable to precision machine tools with limited strokes; however, the compensation effect tends to be less effective for large-stroke machine tools (such as gantry machines) characterized by considerable parameter variability. In such cases, employing online identification and online compensation methods may yield favorable results, although this approach necessitates additional high-performance computing resources and acquisition cards, thereby increasing hardware costs.

A linear motor platform was constructed, detailing the basic configuration and parameters of the system. The parameter identification method for the LuGre friction compensation model was elaborated upon, and the effectiveness of the friction compensation method, along with the robust performance of the designed controller, was validated on the physical platform.

### 5.1 Calculation of target frictional force via motor dynamics

The general mechanical dynamics model for the x-axis of a linear synchronous motor (LSM) servo system is expressed as:

$$M_x \cdot \ddot{y} + C_x \cdot \dot{y} + K_x \cdot y = -F_{rx} + F_x + F_{dis} \tag{23}$$

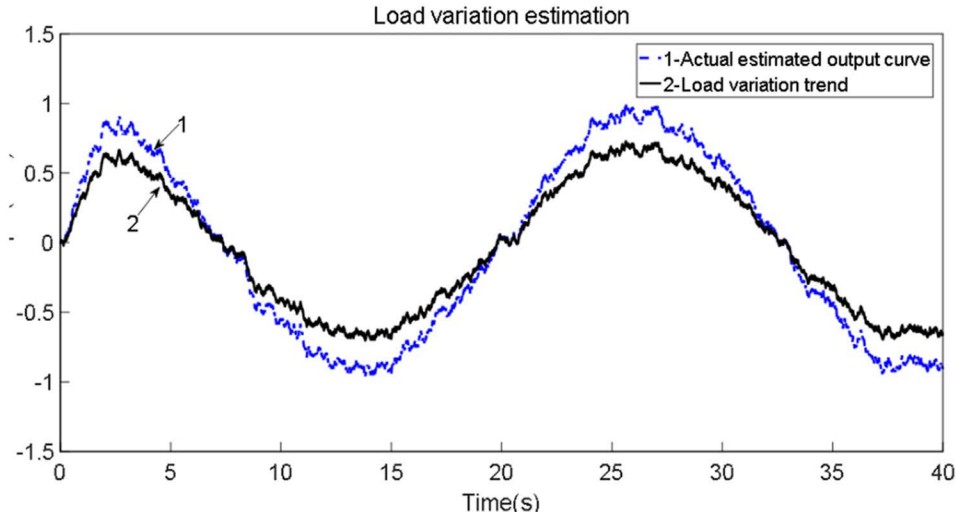

## (a) Load variation estimation

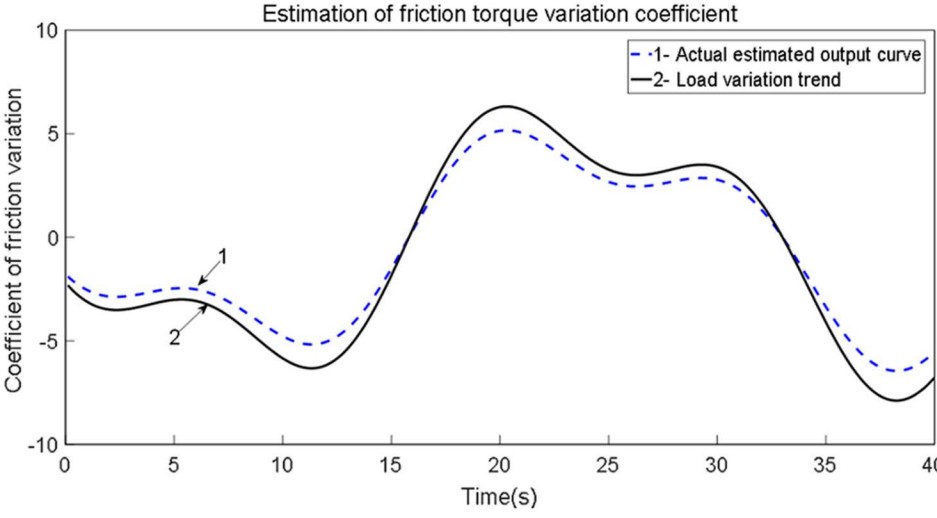

## (b) Estimation of friction torque variation coefficient

**Fig 14. Estimation of load changes.**

Where $y$, $\dot{y}$ and $\ddot{y}$ denote the displacement, velocity, and acceleration of the LSM, respectively. $M_x$, $C_x$ and $K_x$ represent the moving mass, equivalent viscous damping coefficient, and stiffness constant of the X-axis. $F_x = K_F \cdot K_u \cdot u_x$ is the electromagnetic driving force ($K_F$: force constant; $K_u$: amplifier gain; $u_x$: control input), $F_{rx}$ is the friction force, and $F_{dis}$ encompasses unmodeled disturbances and external interference.

By neglecting additional external characteristics of the linear motor, the simplified nonlinear dynamics model of the x-axis reduces to:

$$M_x \cdot \ddot{y} = -F_{rx} + F_x + F_{dis} \tag{24}$$

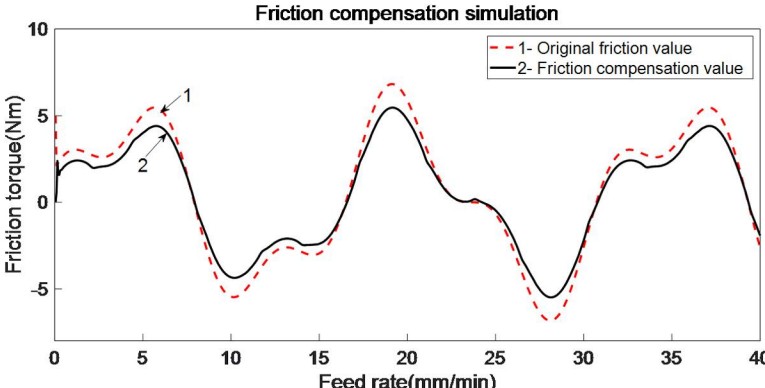

**Fig 15. Friction compensation simulation.**

Here $M$ is the load mass; $F = K \cdot u$ is the electromagnetic driving force ($K = K_F \cdot K_u$: lumped gain), $F_r$ is the frictional force. Equation (24) describes the rigid-body dynamics of the motor as a second-order system with control voltage $u$ as the input and displacement $y$ as the output:

$$M \cdot \ddot{y} = -F_r + K \cdot u + F_{dis} \qquad (25)$$

The LuGre friction model characterizes $F_r$ using stiffness coefficient $\sigma_0$, damping coefficient $\sigma_1$, and average bristle deformation $z$:

$$F_r = \sigma_0 \cdot z + \sigma_1 \cdot \dot{z} + \sigma_2 \cdot \dot{x}(t) \qquad (26)$$

Under pre-sliding conditions (i.e., no macroscopic motion, $z \approx x$ and $\dot{z} \approx \dot{x}(t)$), the friction force simplifies to:

$$F_r \approx \sigma_0 \cdot x + \sigma_1 \cdot \dot{x} + \sigma_2 \cdot \dot{x} \qquad (27)$$

That yields:

$$M \cdot \ddot{x} = -(\sigma_0 \cdot x + \sigma_1 \cdot \dot{x} + \sigma_2 \cdot \dot{x}) + K \cdot u + F_{dis} \qquad (28)$$

Ignoring $F_{dis} = 0$, the simplified equation is described as:

$$M \cdot \ddot{x} = -(\sigma_0 \cdot x + \sigma_1 \cdot \dot{x} + \sigma_2 \cdot \dot{x}) + K \cdot u \qquad (29)$$

Simplify the equation into an expression:

$$M \cdot \ddot{x} + (\sigma_0 \cdot x + \sigma_1 \cdot \dot{x} + \sigma_2 \cdot \dot{x}) = K \cdot u \qquad (30)$$

The input is defined as the control voltage, and the output is the motor displacement $x$. Applying the Laplace transform yields the transfer function:

$$\frac{X_{(s)}}{U_{(s)}} = \frac{K}{Ms^2 + (\sigma_1 + \sigma_2)\, s + \sigma_0} \qquad (31)$$

It is evident that when the system is actuated by electromagnetic driving forces and remains in the pre-sliding regime (i.e., prior to macroscopic motion), its dynamics can be approximated as a linear second-order damped system. Let the natural frequency be denoted as $\omega_n$ and the damping ratio as. The governing equation can then be expressed in the canonical form of a second-order system:

$$\frac{X_{(s)}}{U_{(s)}} = \frac{K}{\sigma_0} \cdot \frac{\omega_n^2}{s^2 + 2\xi \cdot \omega_n \cdot s + \omega_n^2}$$

(32)

Where, the expressions for the natural frequency $\omega_n$ and damping coefficient $\xi$ are described as:

$$\omega_n = \sqrt{\sigma_0/M}$$

(33)

$$\xi = \frac{\sigma_1 + \sigma_2}{2\sqrt{\sigma_0/M}}$$

(34)

From the knowledge of control theory, it can be concluded that:

$$x_{ss} = \frac{K}{\sigma_0}l$$

(35)

$$M_p = e^{-\frac{\varepsilon\pi}{\sqrt{1-\varepsilon^2}}} \times 100\%$$

(36)

Where, $x_{ss}$ is the steady-state value of step response; $M_p$ is the overshoot of the response.

Consequently, under steady-state conditions, a step control voltage input may be introduced to the linear motor system to record its step response. The dynamic parameters of the system can then be systematically identified through analysis of the steady-state displacement and percentage overshoot derived from the transient response profile.

As illustrated in Fig 16, the step response of the stationary motor under a control voltage excitation of $u = 0.1V$ exhibits a steady-state displacement of $X_{ss} = 2.737 \times 10^{-4}$ and a percentage overshoot of $M_p = 43.16\%$. The dynamic parameters

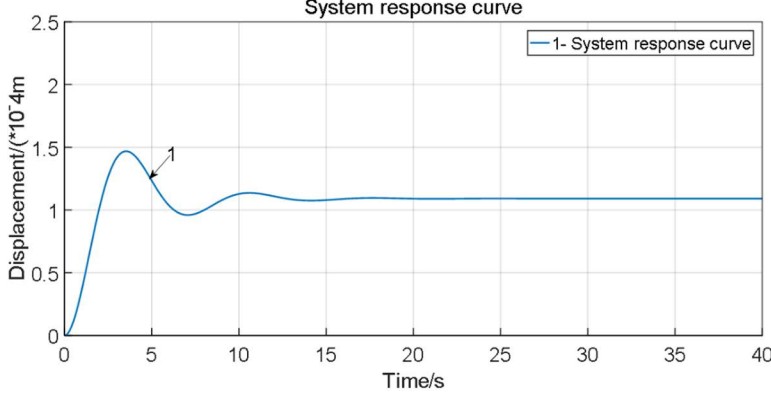

**Fig 16. System response curve.**

of the LuGre friction compensation model are computed as $\sigma_0 = 2421.525 N \cdot m/rad$ and $\sigma_1 = 181.525 N \cdot m \cdot s/rad$. These values quantitatively characterize the pre-sliding stiffness and viscous damping properties of the frictional interface.

## 6 Experiment and results

LuGre friction model parameters are shown in Table 1.

When developing the controller, it is essential to ascertain the parameters of the friction model within the real system to effectively apply the chosen friction model. The outcomes of parameter identification significantly impact the compensation effectiveness of the friction model, thereby influencing the control accuracy of the controller. Consequently, it is imperative to choose suitable parameter identification methods to accurately determine model parameters, thus ensuring the attainment of high-precision compensation.

### 6.1 Sinusoidal signal tracking simulation

To evaluate the influence of bristle stiffness on friction torque, the LuGre model was employed to analyze the tracking accuracy of friction torque and the behavior under micro-displacement and micro-velocity conditions. The experiment was designed using the Matlab Simulink simulation module and subsequently implemented in TwinCAT3 software. The system was simulated and analyzed with the following parameters: simulation inputs set to $v(t) = 4sin(2\pi t)$, a simulation duration of 3 seconds, a fixed-step simulation type with a step size of 0.0001 seconds, and the solver configured to use the ode4 method. The LuGre parameters were simulated and compared separately using both the original and identified parameters. The corresponding time-dependent friction torque curves and identification results are illustrated in Fig 17, which also includes the speed-friction torque curve.

From Fig 17, it is evident that when the friction torque crosses the zero point during upward and downward motion, the tracking error waveform becomes distorted and exhibits a peak phenomenon. This indicates that the stiffness of the bristles significantly influences the friction torque.

### 6.2 Friction compensation experiment

**(1) Experimental study on LuGre model feedforward friction compensation controller.** To mitigate the adverse effects of nonlinear friction on system performance, a feedforward friction compensation controller based on the LuGre model was designed. The control architecture employs a three-loop strategy, where the current loop is embedded within the motor driver, while the speed and position loops are externally implemented. The controller was developed in Simulink and deployed via TwinCAT3 software for real-time execution.

The compensation scheme utilizes feedback speed as the input to the LuGre friction model. To address sampling-period delays inherent in feedback-based compensation, a feedforward friction compensation structure (Fig 18) was adopted. This structure directly injects the estimated friction force into the control input, bypassing latency issues and enhancing system responsiveness.

**Table 1. LuGre friction model parameters.**

| Friction force name | Symbol | Unit | Value |
|---|---|---|---|
| Mane stiffness coefficient | $\sigma_0$ | $(N \cdot m/rad)$ | 242 |
| Mane damping coefficient | $\sigma_1$ | $(N \cdot m \cdot s/rad)$ | 8 |
| Coefficient of viscous friction | $\sigma_2$ | $(\times 10^{-6} N \cdot m \cdot s/rad)$ | 4.82 |
| Static friction force | $F_s$ | N | 1 |
| Stribeck speed | $v_s$ | m/s | 0.001 |
| Coulomb friction force | $F_c$ | N | 1.5 |

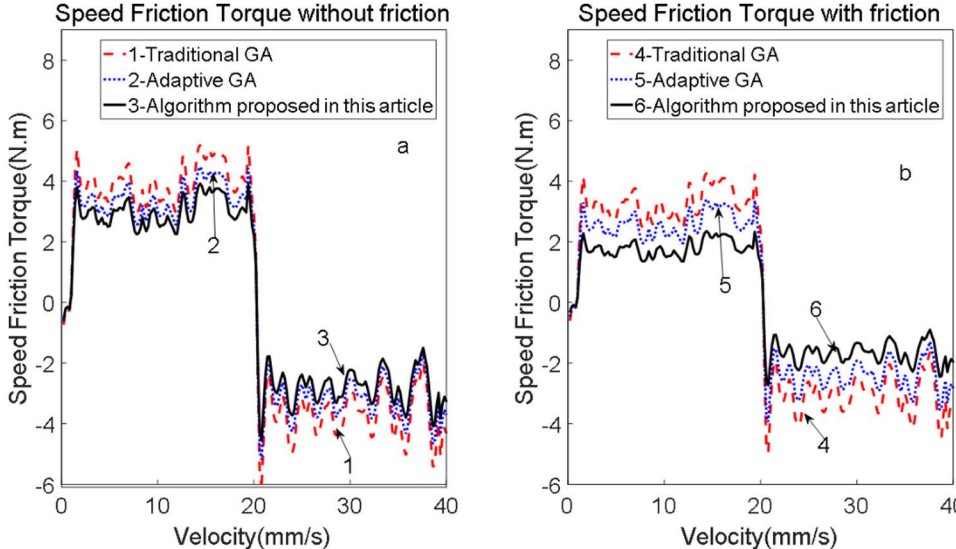

**Fig 17. Comparison results of different speeds and friction torques.**

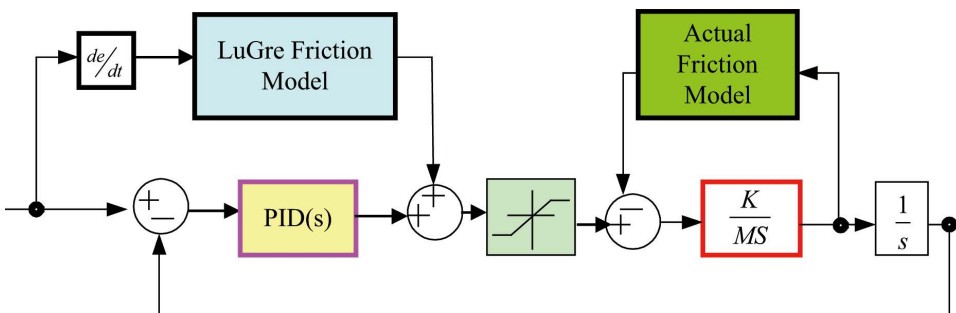

**Fig 18. Feedforward friction compensation model.**

By introducing a sinusoidal signal $x(t) = 15[\cos(\pi t) - \frac{\pi}{6}]$ into both control systems, the signal tracking performance is analyzed. The experiments are divided into two groups based on the presence or absence of feedforward compensation for comparative analysis. The speed tracking curves before and after friction compensation are illustrated in Fig 19.

As illustrated in the speed tracking results, a 'dead zone' is observed during zero-speed transitions. The implementation of friction feedforward compensation significantly mitigates this issue, achieving minimal position tracking error and smooth velocity transitions at the zero-crossing point. As shown in Fig 19, the position tracking error for the fuzzy PID control system with friction feedforward compensation is notably reduced.

The figure demonstrates that with friction compensation, the system maintains stability, effectively balancing and suppressing friction. The position tracking curves before and after the application of friction compensation are illustrated in Fig 20.

The experimental results demonstrate that feedforward friction compensation control based on the LuGre model significantly outperforms traditional PID control in addressing speed tracking imbalances caused by nonlinear friction. This improvement enhances both speed and position tracking accuracy, showcasing greater practical applicability in engineering contexts.

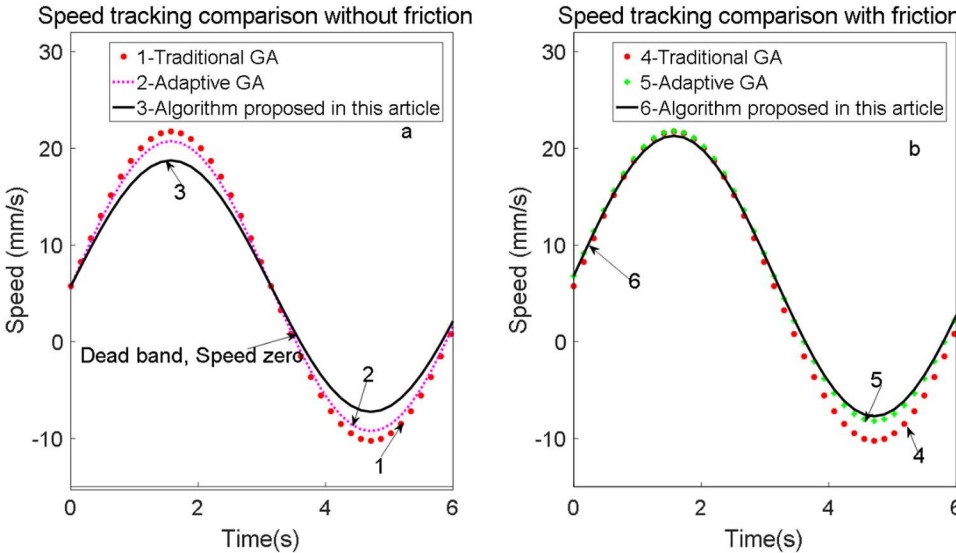

**Fig 19. Speed tracking curve before and after friction compensation.**

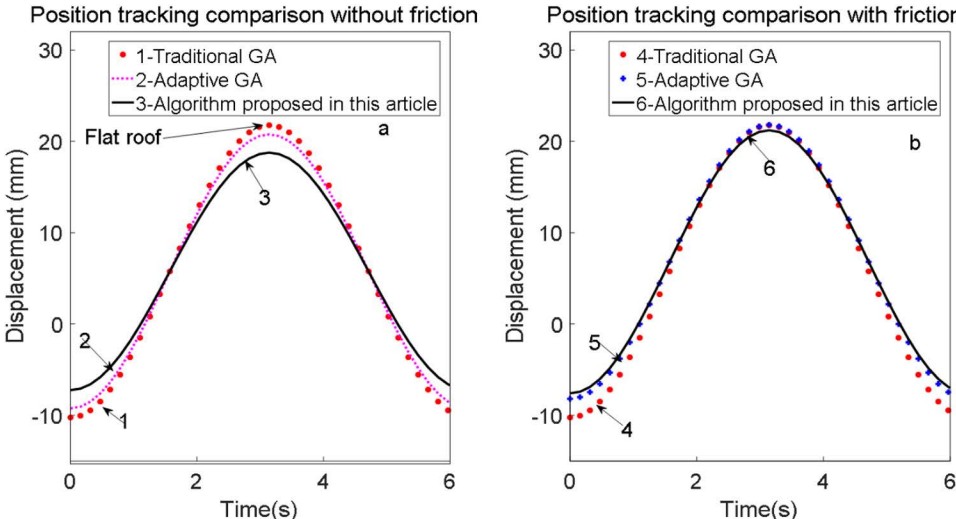

**Fig 20. Position tracking curve before and after friction compensation.**

**(2) Control shear force and control force of identification values.** For the LuGre model, static parameter identification follows conventional methods, while dynamic parameter identification utilizes sensor outputs (displacement or acceleration) and the control shear force as direct inputs. During identification, the control shear force serves as the target approximation value, simplifying the identification of two dynamic parameters.

For dynamic parameter identification, the servo system's displacement (or acceleration) and control shear force outputs are directly used, with the control shear force acting as the target approximation value. Assuming the dynamic parameter $x_d = \omega_2 = [\hat{\sigma}_0 \ \hat{\sigma}_1]$ requires identification, the identification error is defined as:

$$e(x_d, t_i) = u(t_i) - u(x_d, t_i) \tag{37}$$

Where, $u(t_i)$ controls the shear force for the servo system, while $u(x_d, t_i)$ model parameter system identifies the output control force.

$$u(x_d, t_i) = F_f(t) + m\frac{d^2_{x_i}}{dt^2_i} \tag{38}$$

Substituting $z(t) \to z, \dot{z} \to \frac{dz(t)}{d(t)}, \ \dot{x} \to x$ yields equation:

$$F_f(t) = \hat{\sigma}_0 \cdot z + \hat{\sigma}_1 \cdot \dot{z} + \hat{\sigma}_2 \cdot \dot{x} \tag{39}$$

$$\dot{z} = \dot{x} - \frac{\hat{\sigma}_0}{g(\dot{x})}z \tag{40}$$

The objective function is defined as:

$$J = \frac{1}{2}\sum_{i=1}^{N} e^2(y_d, t_i) \tag{41}$$

During the identification of power parameters, the servo system outputs $x(t) = 0.01sin(2.2\pi \cdot t)$, and the corresponding identification results for the control force $u$ and friction force are illustrated in Fig 21. Throughout the identification process, the parameter population search space is defined as $F_c \in [0, 100], F_s \in [0, 100], \ v_s \in [0, 0.1], \sigma_0 \in [0, 1000], \sigma_1 \in [0, 1000], \sigma_2 \in [0, 50]$. It is confirmed that the fuzzy PID control system, integrated with friction feedforward compensation, exhibits both rapid response and steady-state stability. The identification values for friction, control force, and displacement loop are illustrated in Fig 22.

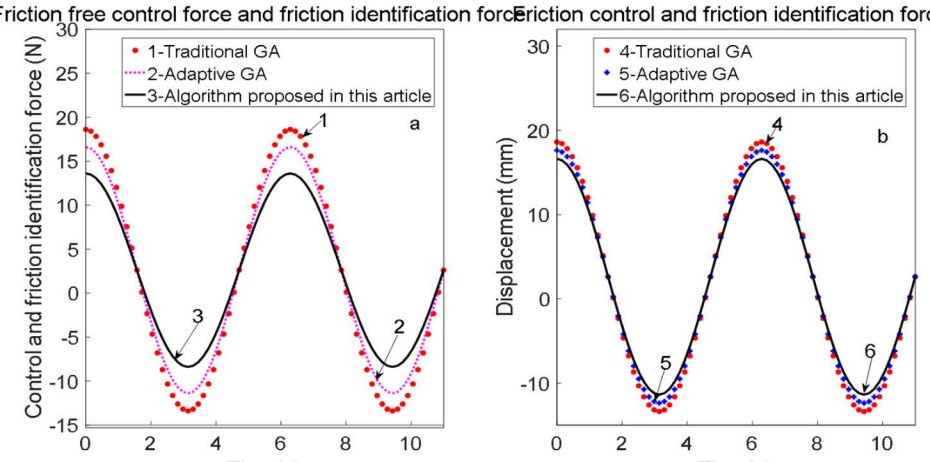

**Fig 21. Control shear force and control force of identification values.**

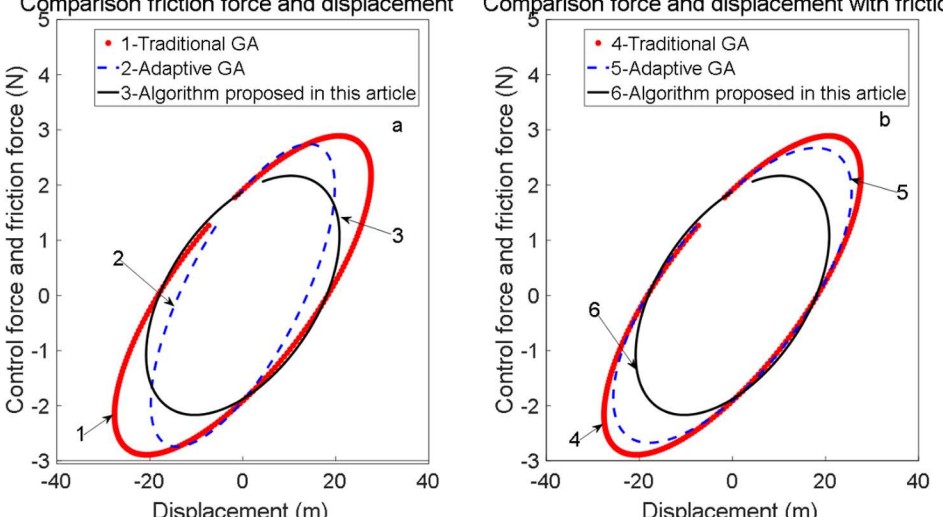

**Fig 22. Friction, control force and displacement loop of identification value.**

The discrepancy between the friction and control force in the two figures is attributed to inertial force; specifically, the control force is equivalent to the sum of friction and inertial force. Subsequently, the LuGre friction compensation model was introduced and compared with the simplified model under the same control law, resulting in a significant reduction in tracking error. This fully demonstrates the efficacy of the LuGre friction compensation model.

From the parameter identification results, traditional genetic algorithms (GA), adaptive genetic algorithms, and the proposed algorithm exhibit high accuracy and fast identification speed in determining the static and dynamic nonlinear parameters of the LuGre friction model, establishing them as ideal tools for nonlinear parameter identification.

Building upon the LuGre friction model and an enhanced adaptive genetic identification algorithm, the findings of this research are anticipated to mitigate the adverse effects of nonlinear friction on high-precision and ultra-low-speed servo systems. Nonetheless, constrained by various factors, the proposed method's performance has been initially corroborated only within a simulation framework. Prospective research endeavors will concentrate on the following domains:

We aim to implement the algorithm delineated in this study within actual high-precision and ultra-low-speed servo systems to ascertain its performance in real-world engineering scenarios, and to further refine the algorithm adaptively to tackle emergent challenges. Although the convergence velocity of the algorithm presented herein surpasses that of the adaptive genetic algorithm, it still lags behind the conventional genetic algorithm. Enhancing the proposed algorithm to accelerate convergence and enhance its suitability for engineering applications remains a pivotal objective for future investigations.

## 7 Summary

(1) In high-precision and ultra-low-speed servo systems, the presence of nonlinear friction significantly impacts both dynamic and static performance. This is primarily evident through phenomena such as low-speed crawling, substantial static errors, and steady-state limit cycles. The basic principles and mathematical representations of the LuGre friction model are introduced in this context. This article presents an enhanced identification method for the LuGre model utilizing a genetic algorithm. Simulation analysis of this improved genetic algorithm demonstrates that, compared to traditional PID control, the incorporation of feedforward friction compensation based on the LuGre model effectively mitigates speed tracking imbalances caused by nonlinear friction. This leads to enhanced speed and position tracking accuracy, thereby offering substantial practical engineering value.

(2) A linear motor platform was established, detailing its fundamental characteristics and parameters. The parameter identification methodology for the LuGre friction compensation model is explained, confirming the effectiveness of the friction compensation technique and the robust performance of the designed controller on the physical platform.

(3) Experimental validation indicates that employing the LuGre friction compensation model substantially enhances the tracking accuracy of linear motor control systems. The improved LuGre genetic algorithm identification controller exhibits commendable control performance, and the synergy between these two approaches facilitates high-precision control outcomes.

## Supporting information

**S1 File. LuGre Model---dataset.**
(ZIP)

## Author contributions

**Conceptualization:** Jingxuan Zhang, Honghong Sun.

**Data curation:** Honghong Sun.

**Formal analysis:** Wanjun Zhang, Feng Zhang, Honghong Sun.

**Investigation:** Jingyan Zhang, Honghong Sun.

**Methodology:** Wanjun Zhang, Jingxuan Zhang.

**Project administration:** Wanjun Zhang, Feng Zhang, Jingyan Zhang.

**Resources:** Jingxuan Zhang, Siyan Zhang, Jingyan Zhang.

**Software:** Jingxuan Zhang, Siyan Zhang, Jingyi Zhang.

**Supervision:** Feng Zhang, Siyan Zhang, Jingyi Zhang.

**Validation:** Wanjun Zhang, Siyan Zhang, Jingyi Zhang, Kristian E Waters.

**Visualization:** Feng Zhang, Jingyi Zhang.

**Writing – original draft:** Hao Ma.

**Writing – review & editing:** Kristian E Waters, Hao Ma.

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
