## [Decision Letter · Decision Letter 0]

22 Dec 2024

PONE-D-24-55538Research and Analysis of an Enhanced Genetic Algorithm Identification Method Based on the LuGre ModelPLOS ONE

Dear Dr. Ma,

Thank you for submitting your manuscript to PLOS ONE. After careful consideration, we feel that it has merit but does not fully meet PLOS ONE’s publication criteria as it currently stands. Therefore, we invite you to submit a revised version of the manuscript that addresses the points raised during the review process.

**ACADEMIC EDITOR: **As per the reviewer's feedback, I am recommending "Major Revision" for this paper. Authors must go through each of the reviewer comments very seriously and make a sincere point-to-point response and improve their paper accordingly. If authors will be able to undertake this task then I will be happy to reconsider my decision. 

We look forward to receiving your revised manuscript.

Kind regards,

Himadri Majumder, Ph.D

Academic Editor

PLOS ONE

3. We note that your Data Availability Statement is currently as follows: [All relevant data are within the manuscript and its Supporting Information files.] Please confirm at this time whether or not your submission contains all raw data required to replicate the results of your study. Authors must share the “minimal data set” for their submission. PLOS defines the minimal data set to consist of the data required to replicate all study findings reported in the article, as well as related metadata and methods (https://journals.plos.org/plosone/s/data-availability#loc-minimal-data-set-definition).

Additional Editor Comments:

As per the reviewer's feedback, I am recommending "Major Revision" for this paper. Authors must go through each of the reviewer comments very seriously and make a sincere point-to-point response and improve their paper accordingly. If authors will be able to undertake this task then I will be happy to reconsider my decision.

Reviewers' comments:

Reviewer's Responses to Questions

**Comments to the Author**

1. Is the manuscript technically sound, and do the data support the conclusions?

Reviewer #1: No

Reviewer #2: Partly

Reviewer #3: Yes

2. Has the statistical analysis been performed appropriately and rigorously? 

Reviewer #1: No

Reviewer #2: I Don't Know

Reviewer #3: Yes

3. Have the authors made all data underlying the findings in their manuscript fully available?

Reviewer #1: No

Reviewer #2: Yes

Reviewer #3: No

4. Is the manuscript presented in an intelligible fashion and written in standard English?

Reviewer #1: No

Reviewer #2: Yes

Reviewer #3: Yes

5. Review Comments to the Author

Reviewer #1: In this paper, an improved genetic algorithm is proposed to identify parameters of the Lugre model, and relevant theoretical derivation is carried out, and the effectiveness of the method is verified through simulation and experiment. However, the theoretical derivation of the paper is confused, the format is different, and the information density of the pictures is low, so the simulated and experimental images cannot fully correspond to the text before and after. Here are some specific suggestions.

1.Formula (1) does not give the definition of all variables;

2.Figure 2 is too sloppy, and it is not combined with the article. It only lists the control block diagram of a certain control method, without using the variables given in this article;

3.The mathematical expression of equation (7) is irregular;

4.The words in Figure 3 overlap each other, and it is not clear how to identify parameters, please modify it carefully, and describe the specific process of parameter recognition in combination with the research object and target parameters of this paper;

5.Please add experiments to prove the effectiveness of the improved genetic algorithm compared with the traditional algorithm;

6.Increase the interpretation of experimental results;

7.Please carefully check the control methods compared in the experiment, and there are many language expression errors in the text, such as : ( 1 ) The upper part of Figure 14 represents the control effect of fuzzy PID, and the lower part represents the control effect of traditional PID ; ( 2 ) The lower part of Figure 15 shows that the fuzzy PID control system combined with friction feedforward compensation is proposed in this paper, which is inconsistent with the theme of this paper.

8.The contribution to method innovation is not significant.

Reviewer #2: - Briefly summarize the improvements made by the proposed friction compensation technique at the end of the abstract. Mention quantitative improvement.

- Instead of writing 'Reference [5]' , 'Reference [7]', write first author name followed by et al.

- Also, define all the abbreviations at their first occurrence in the manuscript e.g. PID should be completely defined at its first occurrence.

- The first paragraph outlining the crucial role of fiction in physical systems could benefit from literature references such as; Adaptive FIT-SMC approach for an anthropomorphic manipulator with robust exact differentiator and neural network-based friction compensation

- Please fix the overwriting of text and symbols in Figure 3.

- Correct the flowchart in Figure 5. In decision block, are there two different conditions being evaluated? If Yes, is it OR or ND between these conditions? Also, the text of the second last block needs to be updated to "Simulate annealing on the new species".

- Please include paper outlines at the end of Section 1 (Introduction)

- Elaborate discussion on friction compensation techniques with reference to literature e.g. 10.1371/journal.pone.0256491 and 10.1371/journal.pone.0258909

- Label Figure 7 so as to convey more useful information.

- What are the units of coefficients in Table 1?

- Include results on Step response to precisely characterise the performance of the proposed approach.

- Update the literature review on LuGre model with notable works such as 'Control of an anthropomorphic manipulator using LuGre friction model - Design and experimental validation'

- Section 5.1 presents the simulation results. Please clearly outline the heading for Experimental Results. Are they mentioned in Section 5.2?

- Also, the results need more critical and conclusive discussion. Moreover, please include limitations of the presented study.

- Please thoroughly proofread the paper for typos and linguistic improvements.

Reviewer #3: Comments to improve:

1. In abstract and also in the introduction there were some phrases that was repeated similarly, it’s better to have a different type of explanation and more summarized in abstract.

2. It is better to also have an explanation on how to calculate the target friction force based on motor dynamic equations.

3. The arrows in Fig.2 could have been shown more understandable.

Strength Points:

1. Having a comparation between suggested controller and another type to display the improvement and efficiency.

2. Mentioning the strength and week points of each method used, in introduction.

3. Good breakdown of parameters to identify.

4. Defining the LuGre equations in full term and not simplifying it.

Week Points:

1. In introduction it is better to focus on evaluated of LuGre model instead of mentioning the history of this method.

2. The equations of LuGre coefficients in Eq.2 could have been introduced separately.

3. At section “Improved Annealing GA” the dictation of LuGre is incomplete.

4. The input of flowchart is incorrect at Fig.3.

5. The error between two plots is considerable at Fig.11.

Questions:

1. What was the case of choosing a liner motor in order to evaluate a nonlinear friction model?

2. In Eq.1 the full formula has a parameter α as the power of e, why in this equation it was determined 2 and no other values?

3. On which contact surface do we have the deflection of bristles?

4. Why the identification matrix does not contain parameter σ_2?

5. In Eq.6 the parameter “k” stands for number of irritations or time step since the parameter z(t) is a time dependent variable?

The up to date references should be added to survey.

[1] P. Moradi, M. H. Korayem, N. Yousefi, “Extended Nonlinear Time-varying Lugre-based Friction Model Identification of Robot Manipulator with SMC Compensation Approach”, Proceedings of the Institution of Mechanical Engineers, Part I: Journal of Systems and Control Engineering 237, no. 2: 207-219, 2023.

[2] P. Moradi, M. H. Korayem, N. Yousefi, “Online Identification and Robust Compensation of Extended Nonlinear Time-varying Friction Model in Robotic Arms”, Journal of Mechanical Science and Technology 37, no. 1: 367-373, Jan 2023.

[3] M. H. Korayem, M. Zakeri, and M. Taheri, “Simulation of Two-Dimensional Nanomanipulation of Particles Based on the HK and LuGre Friction Models”, Arabian Journal for Science and Eng., Vol. 38, No. 6, pp. 1573-1585, February 2013.

[4] M. H. Korayem, A. Hedayat, and S. F. Dehkordi, “Application of frictional contact to extend the functionality of cooperative manipulator chains in moving a common object in the form of closed kinematic chain”, Applied Mathematical Modelling, Vol. 90, Pages 302-326, Feb. 2021.

6. PLOS authors have the option to publish the peer review history of their article (what does this mean? ). If published, this will include your full peer review and any attached files.

**Do you want your identity to be public for this peer review?** For information about this choice, including consent withdrawal, please see our Privacy Policy .

Reviewer #1: No

Reviewer #2: No

Reviewer #3: No

---

## [Author Response · Author response to Decision Letter 1]

22 Mar 2025

Please check the attached file for the response to reviewers

---

## [Decision Letter · Decision Letter 1]

31 Mar 2025

Research and Analysis of an Enhanced Genetic Algorithm Identification Method Based on the LuGre Model

PONE-D-24-55538R1

Dear Dr. Ma,

We’re pleased to inform you that your manuscript has been judged scientifically suitable for publication and will be formally accepted for publication once it meets all outstanding technical requirements.

Kind regards,

Himadri Majumder, Ph.D

Academic Editor

PLOS ONE

Additional Editor Comments (optional):

Reviewers' comments:

Reviewer's Responses to Questions

**Comments to the Author**

1. If the authors have adequately addressed your comments raised in a previous round of review and you feel that this manuscript is now acceptable for publication, you may indicate that here to bypass the “Comments to the Author” section, enter your conflict of interest statement in the “Confidential to Editor” section, and submit your "Accept" recommendation.

Reviewer #1: All comments have been addressed

Reviewer #2: All comments have been addressed

Reviewer #3: All comments have been addressed

2. Is the manuscript technically sound, and do the data support the conclusions?

Reviewer #1: Yes

Reviewer #2: Yes

Reviewer #3: Yes

3. Has the statistical analysis been performed appropriately and rigorously? 

Reviewer #1: Yes

Reviewer #2: I Don't Know

Reviewer #3: N/A

4. Have the authors made all data underlying the findings in their manuscript fully available?

Reviewer #1: Yes

Reviewer #2: Yes

Reviewer #3: No

5. Is the manuscript presented in an intelligible fashion and written in standard English?

Reviewer #1: Yes

Reviewer #2: Yes

Reviewer #3: Yes

6. Review Comments to the Author

Reviewer #1: I have carefully reviewed the revised manuscript, and the author has made the necessary improvements based on the feedback provided. The paper now meets the requirements for publication.

Reviewer #2: The authors have addressed all the comments suggested. The revised version of the paper has been improved significantly and is recommended for acceptance. Please include high-resolution images in the final submission.

Reviewer #3: The authors have made substantial revisions to improve the quality of the paper, according to the comments. All my questions have been solved. I think this version can be accepted.

7. PLOS authors have the option to publish the peer review history of their article (what does this mean? ). If published, this will include your full peer review and any attached files.

**Do you want your identity to be public for this peer review?** For information about this choice, including consent withdrawal, please see our Privacy Policy .

Reviewer #1: No

Reviewer #2: No

Reviewer #3: No

---

## [Editor Report · Acceptance letter]

PONE-D-24-55538R1

PLOS ONE

Dear Dr. Ma,

I'm pleased to inform you that your manuscript has been deemed suitable for publication in PLOS ONE. Congratulations! Your manuscript is now being handed over to our production team.

Kind regards,

on behalf of

Dr. Himadri Majumder

Academic Editor

PLOS ONE